# Characterizing acceptable and appropriate implementation strategies of a biobehavioral survey among men who have sex with men and others assigned male who have sex with men in Zimbabwe

**Lauren E. Parmley**[1]*, **Sophia S. Miller**[1], **Tiffany G. Harris**[1,2], **Owen Mugurungi**[3], **John H. Rogers**[4], **Avi Hakim**[5], **Godfrey Musuka**[6], **Innocent Chingombe**[6], **Munyaradzi Mapingure**[6]

**1** ICAP at Columbia University, New York, New York, United States of America, **2** Department of Epidemiology, Columbia University Mailman School of Public Health, New York, New York, United States of America, **3** Zimbabwe Ministry of Health and Child Care, Harare, Zimbabwe, **4** Division of Global HIV & TB, U.S. Centers for Disease Control and Prevention, Harare, Zimbabwe, **5** Division of Global HIV & TB, U.S. Centers for Disease Control and Prevention, Atlanta, Georgia, United States of America, **6** ICAP at Columbia University, Harare, Zimbabwe

* lparmley@usaid.gov

## Abstract

Key populations including men who have sex with men (MSM), female sex workers, people who inject drugs, transgender persons, and prisoners account for nearly 50% of new HIV infections globally. To inform the HIV response and monitor trends in HIV prevalence and incidence among key populations, countries have increased efforts to implement biobehavioral surveys (BBS) with these groups as part of routine surveillance. Yet the marginalized nature of populations participating in a BBS requires contextually acceptable and appropriate strategies for effective implementation. We conducted a formative assessment to inform the first BBS conducted with MSM and others assigned male who have sex with men (OAMSM) in Zimbabwe, where same-sex sexual behaviors are illegal and highly stigmatized and describe applications of our findings. Qualitative data were collected through four focus groups with 32 MSM/OAMSM and 25 in-depth interviews (15 MSM/OAMSM, 10 service providers/gatekeepers) from December 2018 to January 2019. Rapid assessment techniques were employed including rapid identification of themes from audio recordings and review of detailed field notes and memos to identify key themes. Findings from this assessment included contextually relevant considerations including behaviors and terminology to avoid when working with MSM/OAMSM in Zimbabwe, appropriate compensation amounts for survey participation, proposed data collection sites, and differences in sexual openness, marital status, and networks among younger and older MSM/OAMSM. Participants also reported strong network ties suggesting respondent-driven sampling—a peer chain referral approach—to be an appropriate recruitment method in this context. Taken together, these findings highlighted key considerations and strategies for implementation to ensure the subsequent BBS in Zimbabwe was both acceptable and appropriate. These

**Data Availability Statement:** Data are available upon request. Participant consent forms specified that only study investigators would have access to study records. Additionally, data may have elements of personal identification and highly sensitive information and given the criminalized nature of same sex sexual behaviors in Zimbabwe, requests for access to data should be sent for review to Dr. Brian Moyo (Epidemiologist, AIDS and TB Programme at MOHCC) at boyobk1@gmail.com. Individuals seeking access to data will need appropriate IRB and institutional (ICAP at Columbia University, MOHCC, CDC) leadership approval, which will be facilitated by Dr. Brian Moyo.

**Funding:** This project was supported by the President's Emergency Plan for AIDS Relief through the CDC under the cooperative agreement #NU2GGH001939. The findings and conclusions in this manuscript are those of the authors and do not necessarily represent the official position of the funding agencies. ICAP at Columbia University was the recipient of this award. TGH served as principal investigator for this award and was partially funded under this award. LEP, SMS, TGH, GM, IC, and MM were also funded under this award. OM, JHR, and AH were not funded by this award. JHR and AH are CDC employees; they participated in study design and reviewed the manuscript prior to submission.

**Competing interests:** GM is an editor with PLOS ONE and PLOS GLOBAL HEALTH. All other authors of this manuscript have no competing interests to declare. This does not alter our adherence to PLOS ONE policies on sharing data and materials.

results and applications of these results are important for informing surveillance efforts and broader HIV-related engagement efforts among MSM/OAMSM in Zimbabwe as well as in other contextually similar countries in Southern Africa.

## Introduction

Key populations—populations disproportionately affected by HIV including men who have sex with men (MSM), female sex workers (FSW), people who inject drugs, transgender persons, and prisoners—account for nearly 50% of new HIV infections globally [1]. While HIV estimates among MSM vary by country, aggregate regional estimates from sub-Saharan Africa, where available, suggest the HIV prevalence among MSM is 17.9%, more than threefold that of the general population in the region [2]. Contributing to this burden are structural factors including stigma and criminalization of same-sex sexual behaviors and access to health and social services, as well as biological factors affecting HIV transmission risk through anal sex [2–5].

Data on the HIV burden among key populations in Zimbabwe are consistent with findings throughout sub-Saharan Africa. More than half of FSW and nearly a third of prisoners are living with HIV according to survey estimates [6, 7]. Unlike FSW and prisoners, there is a dearth of population estimates about MSM due in part to the cultural and legal context which shuns and criminalizes same-sex sexual behaviors in Zimbabwe [8]. Sample and programmatic estimates of HIV prevalence among MSM range from 23.5%-31.0% compared to 12.0% among the general adult male population in Zimbabwe [7, 9]. However, data among MSM have limitations as they only reflect results from small convenience samples or are limited to those who sought services.

Biobehavioral surveys (BBS), through respondent-driven sampling (RDS)—a chain referral approach which accounts for sampling bias through weighting—provide a means of approximating population estimates of key demographic, behavioral, and health outcome data, including HIV and sexually transmitted infection (STI) prevalence, for key populations [10]. BBS data are used to inform the HIV response and tailor country and donor resources. As a result, countries are increasingly implementing BBS among key population groups and on a routine basis for surveillance purposes, with global guidance to implement BBS every three years [11]. Given the marginalized nature of populations participating in a BBS, effective implementation of BBS or similar surveys requires acceptable and appropriate implementation strategies for key populations identified through formative research [11].

In Zimbabwe, where no previous BBS using RDS had been conducted among MSM or others assigned male who have sex with men (OAMSM) at the time, ensuring contextually relevant implementation was particularly important to ensure BBS success. To identify acceptable and appropriate implementation strategies to inform a BBS with MSM/OAMSM in two urban areas of Zimbabwe, we conducted a formative assessment with MSM/OAMSM and other key informants. Results from this formative assessment informed the subsequent BBS in Zimbabwe and can inform future BBS implementation and surveillance efforts as well as broader HIV-related engagement efforts among MSM/OAMSM in Zimbabwe and in other contextually similar settings.

## Methods

From December 2018 to January 2019, qualitative data were collected in Harare and Bulawayo, Zimbabwe, where the BBS later took place from March to July 2019. As part of the formative

assessment, we conducted in-depth interviews (IDIs) with 15 MSM/OAMSM and 10 other key informants and four focus groups with 8 MSM/OAMSM per focus group across both cities. IDI and focus group methods were selected to complement one another to explore sensitive issues in-depth and understand the social context and community dynamics. MSM/OAMSM were eligible to participate if they were assigned male sex at birth, had anal or oral sex with a man in the past 12 months, were 18 years or older, resided in Harare or Bulawayo for at least one month, and spoke English/Ndebele/Shona. While cis-gender MSM were the original target population for the formative assessment, eligibility criteria were specific to male sex assigned at birth (for both the formative assessment and subsequent BBS), and individuals were eligible to participate irrespective of gender identity. Eligibility criteria were the same for IDIs and focus groups, and those who were eligible were offered the choice to participate in either an IDI or focus group.

Key informants included MSM/OAMSM healthcare service providers (n = 5) and "gate-keepers", an owner/employee of a local business that catered to MSM/OAMSM, male or female sex worker who interacted frequently with members of the MSM/OAMSM community, or someone with special access to MSM/OAMSM (e.g., trusted by the MSM/OAMSM community) who could speak to the unique HIV prevention, care, and treatment needs of MSM/OAMSM or the social dynamics of the community (n = 5). Key informants were eligible to participate if they were 18 years or older, resided in Harare or Bulawayo for at least one month, spoke English/Ndebele/Shona, and were currently employed ($\geq$6 months in their current position) by a non-governmental organization providing HIV services to MSM/OAMSM (healthcare service provider) or were a "gatekeeper" for the MSM/OAMSM community as described above.

Participants were purposively recruited with support from local non-governmental organizations providing services to the lesbian, gay, bisexual, transgender, and intersex (LGBTI) communities in Harare and Bulawayo and through snowball sampling. To ensure a diverse sample, MSM/OAMSM participants were recruited across key variants including age, marital status, occupation, and other demographic characteristics. Once potential participants were identified, eligibility was assessed. All eligible and interested individuals provided verbal informed consent prior to participation and received compensation of 5 USD for travel and time.

Before commencing the interview/focus group, participants were provided with a description of the proposed BBS, including information on general procedures and approximate interview time. Semi-structured interview (S1 Text) and focus group guides (S2 Text) were used to obtain information about general topics of interest around survey logistics (e.g., identification of MSM/OAMSM community stakeholders and local leaders, networks within the MSM/OAMSM community, acceptance of HIV/STI testing, survey sites, feedback on the appropriate compensation amount, and how the survey should be marketed), as well as discussion of the characteristics of the MSM/OAMSM population and use of (including barriers to) HIV/STI prevention, care, and treatment services. Key informant interview guides (S3, S4 Texts) included additional questions on participants' professional experience, characteristics of their affiliated organization, and types of services provided to MSM/OAMSM (for healthcare service providers) as well as on interactions and engagement with MSM/OAMSM, including ways in which key informants support and protect MSM/OAMSM.

Focus groups and IDIs were moderated by a facilitator/interviewer and detailed field and expanded notes were captured via a notetaker. Interviewers were trained researchers with a bachelor's degree or higher education and were both male and female. Prior to data collection, interviewers underwent a five-day training workshop which included sensitization training on key populations, led by staff from local LGBTI organizations, human subjects research and responsible conduct of research, and qualitative research methods.

Focus groups were held separately for younger (18–29 years) and older (30 years or older) MSM/OAMSM in each city to promote open discussion. All focus groups were audio recorded and recordings were destroyed post-analysis for purposes of confidentiality. IDIs and focus groups were conducted in English, Ndebele, or Shona, according to participant's preference.

Rapid assessment techniques including rapid identification of themes from audio recordings and review of detailed field and expanded notes and reflective and analytic memos were employed as analytic approaches to identify emergent themes and facilitate across-case comparison [12–14]. Data analysis was an ongoing and iterative process that began with the first interview/focus group and continued until all data were collected. Key themes were developed deductively from IDI and focus group guides which focused on generating specific implementation considerations for the subsequent survey and inductively through consultation with interviewers during debrief sessions, rapid assessment techniques, and analytic memoing [12–14]. Deductive themes included: survey participation, biological testing procedures, compensation, and recruitment and data collection methods. Inductive themes included: socio-cultural considerations and interviewing approaches, including importance of confidentiality.

Ethical approvals, including approvals to conduct verbal consent, were received from the Columbia University Institutional Review Board (AAAR8950) and the Medical Research Council of Zimbabwe (MRCZ/A/2156). The protocol was also reviewed in accordance with the US Centers for Disease Control and Prevention (CDC) (2018–444) human research protection procedures and was determined to be research, but CDC investigators did not interact with human subjects or have access to identifiable data or specimens for research purposes. As a signed consent form would have been the only record linking participants to the research and the principal risk of the research was potential harm resulting from a breach of confidentiality, participants provided verbal consent which was documented on a paper consent form by an interviewer's signature. Additional information regarding the ethical, cultural, and scientific considerations specific to inclusivity in global research is included as Supporting Information (S5 Text).

## Results

### Participant characteristics

All potential participants who were screened and found to be eligible agreed to take part in the assessment and provided informed consent. A total of 15 MSM/OAMSM and 10 other key informants participated in IDIs and 32 MSM/OAMSM participated in focus groups. Key informant participants who provided services to MSM/OAMSM had provided healthcare services for a median of 2 years and saw a median of 10 MSM/OAMSM patients per week (Table 1). Healthcare service providers worked in public and private facilities, and included peer mobilizers, nurses, and doctors. Other key informants included sex workers and owners of bars frequented by MSM/OAMSM. Most MSM/OAMSM participants had never been married. Median age of MSM/OAMSM IDI participants was 25 years compared to 30 years for focus groups participants. Other demographic characteristics of MSM/OAMSM and key informant participants are described in Table 1.

### Acceptability

**Survey participation.**   MSM/OAMSM and key informants indicated that MSM/OAMSM would be eager to participate in the BBS. Most MSM/OAMSM participants expressed a strong interest in participation, believing that the survey would promote HIV testing and treatment among MSM/OAMSM who would otherwise be disinclined to access services due to stigma and discrimination.

**Table 1. Demographic characteristics of formative assessment participants.**

| Characteristic | MSM/OAMSM | | Other key informants | |
|---|---|---|---|---|
| | FGD participants (n = 32) | IDI participants (n = 15) | Gatekeepers (n = 5) | Service Providers (n = 5) |
| Age in years, median (range) | 30 (18–63) | 25 (19–34) | 33 (26–45) | 36 (25–46) |
| City | | | | |
| Harare | 16 (50%) | 7 (47%) | 2 (40%) | 2 (40%) |
| Bulawayo | 16 (50%) | 8 (53%) | 3 (60%) | 3 (60%) |
| Marital status | | | | |
| Living together | 0 (0%) | 2 (13%) | - | - |
| Married to a man | 1 (3%) | 2 (13%) | - | - |
| Married to a woman | 2 (6%) | 0 (0%) | - | - |
| Never married | 26 (81%) | 10 (67%) | - | - |
| Divorced or separated | 3 (9%) | 1 (7%) | - | - |
| Highest-level of education completed | | | | |
| Primary | 1 (3%) | 0 (0%) | - | - |
| Secondary | 18 (56%) | 9 (60%) | - | - |
| Tertiary | 13 (41%) | 6 (40%) | - | - |
| Employed in last 12 months, n | | | | |
| Yes | 14 (44%) | 11 (73%) | 5 (100%) | 5 (100%) |
| No | 17 (53%) | 4 (27%) | 0 (0%) | 0 (0%) |
| Pensioner | 1 (3%) | 0 (0%) | 0 (0%) | 0 (0%) |
| Monthly salary in USD, median (range) | 245 (45–6800) | 300 (50–600) | - | - |
| Duration providing services in years, median (range) | - | - | - | 2 (0.5–4) |
| # of patients seen per week, median (range) | - | - | - | 50 (20–300) |
| # of MSM/OAMSM patients seen per week, median (range) | - | - | - | 10 (5–50) |
| # of MSM/OAMSM served by organization, median (range) | - | - | - | 300 (60–1200) |

*"I am very interested in joining the survey and cannot wait. I think it is a smart move to draw MSM closer to accessing health services." (Older MSM/OAMSM focus group, Bulawayo)*

Additionally, MSM/OAMSM and key informants noted that the BBS presented an opportunity to raise awareness about MSM/OAMSM, a historically marginalized population in Zimbabwe. Participants indicated that the survey would bring recognition to MSM/OAMSM in Zimbabwe and provide a platform for MSM/OAMSM to express their experiences, sexuality, and healthcare needs.

*"I think in terms of them [MSM/OAMSM] participating it would not be a challenge for them because they really feel and want to be recognized. They really want to be part of some of the things that are happening, they personally feel they are being shut out, their voices are not heard, so I think when the research is being introduced, they should work with influential people first so that any fears that MSM may have are addressed." (Key informant, Bulawayo, Age 25)*

**Biological testing procedures.** Many participants believed that HIV testing via venous blood draw would be acceptable among MSM/OAMSM. Service providers indicated that MSM/OAMSM were not averse to HIV testing, and many MSM/OAMSM participants described prior experience with HIV testing in other research or clinical settings. MSM/

OAMSM and key informant participants also believed that the benefit of receiving an HIV test and knowing one's status would encourage survey participation.

*"I think [testing is] a very good idea because some of us have got some partners that are reluctant to come for things like HIV testing and so on, so if you can start this initiative, there will be more LGBTI members who will come forward to know their HIV status. . ." (Older MSM/ OAMSM focus group, Harare)*

*"It's a good thing [HIV testing] if we want to achieve ending AIDS so those who test positive will be commenced on ART and those who are negative can be put on PrEP. . .yes they will be willing as some do not know their status." (MSM/OAMSM IDI, Bulawayo, Age not provided)*

Among the few participants who described potential challenges related to HIV testing for their peers such as perceived stigma and/or fear of an HIV-positive result, most reported that HIV testing would be acceptable if conducted by professional and MSM/OAMSM-sensitized staff with appropriate pre- and post-test counselling and assurance of confidentiality.

*"I will be comfortable [testing for HIV] with a friendly and professional nurse." (Older MSM/ OAMSM focus group, Harare)*

*"The nurses just have to be normal, the same way they act when treating any other person who is not an MSM." (MSM/OAMSM IDI, Harare, Age 22)*

### Appropriateness

**Socio-cultural considerations.** Participants described it was common for their MSM/ OAMSM peers to have romantic relationships with women, with many MSM/OAMSM having girlfriends or wives. While some MSM/OAMSM were described to be bisexual, others engaged in heterosexual relationships as a means of hiding their sexual orientation. MSM/OAMSM participants indicated bisexual MSM/OAMSM and closeted MSM/OAMSM engaging in romantic relationships with women would most benefit from HIV prevention education.

*"We do have those we call the 'after 9s'. They are straight by day and then MSM by night. They don't want to be seen interacting with MSM during the day to avoid questions. They only come [out] at night. . .We also have some MSM found [on] the streets because they do not want to be known, [they] may wait for MSM outside the popular bars and hangout places in the street then interact with those coming from the bars from outside" (MSM/OAMSM IDI, Harare, Age 26)*

*"There are men who have wives at home but also have sex with us. They are many." (MSM/ OAMSM, IDI, Harare, Age 22)*

*"Bisexual men and those that are in hiding should be targeted for health care services. . .Some are ignorant on health issues and risks. Some refuse to use condoms and will bluntly ask why a condom, how can I contract HIV from the anus? Bisexual men spread HIV a lot as they are hiding, [they] are scared and do not want to talk about health issues." (Older MSM/OAMSM focus group, Bulawayo)*

Participants reported that younger MSM/OAMSM were more likely to openly self-identify as gay or homosexual, while older MSM/OAMSM were more discrete about their sexual orientation and behaviors due to ostracization in the past and/or fear of losing one's social standing.

Participants reported that older MSM/OAMSM often had wives or live-in girlfriends, including lesbian women, to conceal their sexual orientation, and were less likely to engage with key population-friendly organizations compared to younger MSM/OAMSM.

*"What is also interesting is that you find that young MSMs are more willing to come out and self-identify than the older MSM and its something that we can explain. The older MSM. . .in the past experienced a lot of trauma because of the social ostracization and the politics of the day that did not encourage MSM to come out." (Key informant, Harare, Age 46)*

*"Some older MSM will co-habit with lesbians and live together as husband and wife but, in fact, they will be living in separate rooms while living their separate gay and lesbian lives." (Younger MSM/OAMSM focus group, Harare)*

*"There are those MSM that are over the age of 30, they have fears associated with losing their employment, losing their status in society, family ties and even property. So older MSM tend to go into their shells once they get to ages of around 30." (Key informant, Harare, Age 45)*

Participants indicated that the hard-to-reach nature of older MSM/OAMSM may impede recruitment in the BBS. To recruit older MSM/OAMSM, MSM/OAMSM emphasized the importance of collaboration with friendly, professional, and approachable peer educators who were trusted to keep personal information confidential.

*"He [peer educator] really protects the community and also helps in mobilization. So, I see that this person has excellent inter-personal skills." (Older MSM/OAMSM focus group, Harare)*

MSM/OAMSM and key informants also noted the criminalization and stigmatization of same-sex sexual behaviors in Zimbabwe as relevant socio-cultural considerations. Several MSM/OAMSM participants described previous experiences in which they had been subjected to stigma or physical violence while seeking health services from public facilities or accessing other services from key population organizations.

*"I was once humiliated and harassed at one health institution. . . when I had gone for an HIV test. Upon realizing that I was an MSM, the healthcare worker called her colleagues into the consultation room and when I was leaving the premise I was violently attacked by strangers." (MSM/OAMSM IDI, Harare, Age 30)*

*"When you go and tell them that you have an STI they [healthcare workers] do not attend to you. Even at [facility] they will not attend to you but may call the police" (MSM/OAMSM IDI, Bulawayo, Age 23)*

*"I was stigmatized when I went for HIV testing at a facility which I will not mention. During the pre-counselling session, the counsellor asked if I was married and I said I am single, I do not have a girlfriend, I was asked if I have been tested before and I said yes, then I was asked why I want to get tested if I am single, then I said I have a boyfriend. The counselor was shocked and asked how I can have a boyfriend when I am a man, she questioned me in such a way that I felt violated" (MSM/OAMSM IDI, Harare, Age not provided)*

**Interviewing approaches.**   To feel comfortable discussing sensitive information, such as sexual behavior and drug use, MSM/OAMSM emphasized the need for study staff to be

sensitized toward MSM/OAMSM and trained to avoid judgmental behavior. Although MSM/OAMSM participants described prior encounters in which they experienced discrimination in the healthcare system, most indicated that they would feel comfortable answering questions about sensitive topics if BBS interviewers were generally friendly and non-judgmental toward key populations.

*"Your friendliness, I mean your approach. We are hoping you are the ones who will come next time. If you continue with your friendliness, accommodating us, I think we will definitely tell others in our community that there is such and such a survey and there are friendly people conducting it. This will be a boost. You would have built a foundation. So please continue with your love." (Older MSM/OAMSM focus group, Bulawayo)*

*"If there is suspicion of maybe any uncomfortable capability like in the people who are doing the survey. . .if I am just uncomfortable with the faces of the people. . .no I won't join it". (MSM/OAMSM IDI, Bulawayo, Age 23)*

MSM/OAMSM were sensitive toward the terminology used by future survey staff. Participants identified derogatory terms to avoid when describing MSM/OAMSM, including the derogatory translations for a man who engages in same-sex sexual behaviors in Shona and Ndebele, "Ngochani" and "Stabane", respectively. MSM/OAMSM participants described these words to be offensive, like calling someone a pig or dog.

*"If anyone who is not an MSM calls me 'ngochani', I will fight that person." (Younger MSM/OAMSM focus group, Harare)*

Participants had mixed opinions about the preferred gender of BBS interviewers. Many MSM/OAMSM felt more comfortable being interviewed by a woman, while some wished to be interviewed by another man. Other participants indicated that the gender of the interviewer did not matter if the interviewer acted in a professional manner and was sensitized toward MSM/OAMSM.

*"If I am interviewed by a male person I won't feel as comfortable. My blood will boil. . .our weakness is men, that's our major weakness. Women have a different approach when talking to us, if it were a man, I would be shy and start blushing. Women are easier." (Older MSM/OAMSM focus group, Bulawayo)*

*"[I] would want to talk to someone above 25 [years old] and who has been taught about MSM, someone who is friendly and understanding, mature. Men and women would both be comfortable." (MSM/OAMSM IDI, Bulawayo, Age 29)*

Most MSM/OAMSM, across both age groups, preferred to be interviewed by study staff rather than participating in self-administered interviews via tablet. Although some participants expressed concerns about their ability to discuss sensitive topics with interviewers, many MSM/OAMSM indicated that human interaction with an interviewer would allow them to be more honest in their responses and seek clarification when needed.

*"We want an interviewer, not a computer you cannot speak to or converse with." (Older MSM/OAMSM focus group, Bulawayo)*

*"When I am being interviewed by a person, I am able to follow up on issues I do not understand." (Younger MSM/OAMSM focus group, Harare)"A tablet does not offer an opportunity*

*for interaction and you cannot see emotions to enable trust between participant and project people." (Older MSM/OAMSM focus group, Bulawayo)*

MSM/OAMSM expressed concerns about having their sexual orientation and personal information disclosed. Participants emphasized the need for confidentiality to feel comfortable participating in the BBS; once assured that information would be kept private, MSM/OAMSM expressed willingness to participate in the BBS. MSM/OAMSM and key informants suggested that study staff explain BBS confidentiality practices to participants thoroughly before the questionnaire is administered.

*". . .There are some [MSM/OAMSM] who are weary of these surveys. It also depends who is carrying out the survey, they are afraid of being followed up later." (Key informant, Harare, Age 45)*

"I would be comfortable with [interviewing] because I know that all the information will be treated in strict confidence." (Older MSM/OAMSM focus group, Harare)

**Compensation.**   MSM/OAMSM participants indicated that transportation compensation was essential for participation in the BBS, given the fluctuating economic situation in Zimbabwe. MSM/OAMSM reported that many of their peers were unemployed and would not travel potentially long distances to data collection sites without compensation. When probed on appropriate compensation amounts, some MSM/OAMSM proposed higher compensation figures of up to 20 USD. However, most MSM/OAMSM indicated that an amount of approximately 5 USD would be appropriate compensation for transport and time, with some MSM/OAMSM noting that this amount was similar to or greater than compensation they had received for participation in public health programs.

*"MSM will be prepared to participate in the survey especially if you give them transport money." (MSM/OAMSM IDI, Harare, Age 22)*

*"[A non-governmental organization] has been bringing their truck to our drinking and socializing areas to do HIV and STI screening and they have been giving 2 USD to every MSM who agrees to have a test. So with your [survey], for 5 USD they will definitely come." (MSM/OAMSM IDI, Harare, Age 30)*

**Recruitment and data collection methods.**   Most MSM/OAMSM and key informants felt that RDS, a peer-to-peer recruitment method that begins with purposively selected seeds, would be an acceptable recruitment approach for the BBS. Sexual contact between men is criminalized and often stigmatized in Zimbabwe; MSM/OAMSM were therefore concerned about their sexuality being disclosed to family and community members and it was reported that recruiting less visible MSM/OAMSM networks may be difficult. Given concerns of confidentiality, participants indicated that the use of RDS to recruit participants via trusted friends or peers within the MSM/OAMSM community would be most appropriate for engaging these networks.

*"If people in my community get to know that I am gay, they may hate me because only a few people understand it." (MSM/OAMSM IDI, Bulawayo, Age 24)*

*"Once you get into the MSM network, you are able to identify quite a reasonable number through the networks, but this can only be done if you are able to penetrate their networks." (Key informant, Harare, Age 46)*

Participants described MSM/OAMSM in Zimbabwe to be highly connected and reported that MSM/OAMSM would be eager to recruit their peers. MSM/OAMSM indicated leaders in their community who were outspoken about LGBTI rights and well respected by their peers would be appropriate candidates for recruitment seeds—purposively selected participants who recruit their peers through coupon distribution—as part of RDS.

*"I will obviously accept a coupon because I would have received it from the same type of person, someone who is like me." (Older MSM/OAMSM focus group, Harare)*

*"Yes, there are a lot of influential MSMs who can act as seeds, these include Disk Jockeys, senior MSMs, waiters and bartenders, and employees of MSM-friendly organization." (Key informant, Bulawayo, Age 44)*

When probed on coupon design and information, MSM/OAMSM participants indicated coupons should include the survey name, contact information, survey location and times, and services to be offered. Several participants indicated coupons should depict a logo for MSM/OAMSM who may be illiterate. While most participants preferred coupons not to be branded MSM/OAMSM or LGBTI, some preferred coupons depict a rainbow flag, a known symbol of the LGBTI community.

Though participants expressed mixed views about optimal data collection locations, there was agreement that the space should be private and secure. Many participants indicated that key population-friendly organizations were preferred data collection sites. Although some MSM/OAMSM and key informants reported that older MSM/OAMSM would not be comfortable participating in the survey if data collection were conducted at a location that was publicly known to serve key populations, these sites were generally seen as private, safe spaces for MSM/OAMSM.

*"We have gone to workshops in hotels for example and we have seen people we do not know coming to join us, and they want to spy, so we'd prefer a private and safe space." (Older MSM/OAMSM focus group, Bulawayo)*

*"If you are doing that then it should be at this place we are in, a place where somebody feels safe, because from here I feel safe. It's not just any place like a clinic where people will be looking at you awkwardly. So, I think a venue which is safe, which accommodates us as humans." (MSM/OAMSM IDI, Harare, Age 22)*

*"If it was another location other than [non-governmental organization], I would not be having this discussion with you." (MSM/OAMSM IDI, Harare, Age 19)*

## Applications of formative assessment findings

In this formative assessment, findings informed all aspects of the subsequent BBS, including participant recruitment, staff training, standard operating procedures, and implementation. Both focus group and IDI data emphasized participant privacy and confidentially of the information generated from the survey. These concerns were addressed in the BBS protocol and standard operating procedures, including through confidentiality agreements from staff and

ensuring strict data security and data management procedures. Male and female clinical and non-clinical staff were hired based on participant feedback and a compensation amount of 5 USD, seen to be fair and consistent with participant compensation used in similar surveys according to formative participants, was provided to survey participants.

Socio-cultural sensitivities described by participants, including concerns related to the criminalization and stigmatization of same-sex sexual behavior in Zimbabwe, were also considered during planning and implementation of the BBS. BBS data collection staff underwent key population sensitization training, described by participants to be necessary for survey success. Key population organization's involvement during the training of BBS data collectors extended beyond facilitating sensitization training and included cognitive testing and roleplay of survey questionnaires and procedures, input on participant coupons, review of data collection tools and logs, and discussion of ethical principles in research. To ensure potential participants felt comfortable participating in the BBS within the current social and legal context, key population-friendly data collection sites as well as non-key population-identified sites were established in each city.

Other socio-cultural considerations such as behavior and terminology that should/should not be used by staff, as well as ways to recruit older MSM/OAMSM, were incorporated into training materials and informed seed selection for RDS. Specifically, younger and older MSM/OAMSM were selected as initial seeds to reach networks of different ages and openness in sexual orientation; additional seeds were added during the BBS to recruit under-represented subpopulations, including older MSM/OAMSM [15]. Within five months, target BBS sample size was reached with almost 20% of the combined sample aged 35 years or older [15]. Nearly all (Harare: 97%; Bulawayo: >99%) BBS participants consented to biomarker testing, consistent with findings from the formative assessment that HIV testing was acceptable if conducted with sensitized staff and appropriate pre- and post-test counseling [15].

An operational challenge for the BBS was Zimbabwe's severe hyper-inflation, which drastically worsened during survey data collection. Additionally, toward the end of the BBS, Zimbabwe implemented a water rationing program due to acute water shortages, limiting water to once per week for some residents, and rationing of electricity with power cuts of up to 18 hours per day. While the fluctuating socioeconomic situation was explored to some degree during the formative assessment as part of compensation-related probes, formative assessment tools did not explore instability considerations in-depth, limiting our ability to fully anticipate implementation strategies in the context of an uncertain operating environment.

## Discussion

This formative assessment characterized acceptable and appropriate implementation strategies for conducting a BBS with MSM/OAMSM in Zimbabwe where same-sex sexual behaviors are illegal and highly stigmatized. While formative assessments are included as a key component to BBS implementation in the World Health Organization Biobehavioral Survey Guidelines for Populations at Risk [11], published results from and applications of formative assessments are often only disseminated via country reports which are neither peer-reviewed nor indexed in publicly available databases. Findings from this assessment informed the subsequent BBS and can continue to inform future BBS implementation among MSM/OAMSM in urban Zimbabwe, as well as in other contextually similar countries in Southern Africa.

Existing data highlight ways in which formative results can inform RDS surveys among key populations [16]. Johnston and colleagues explored how, when formative results are overlooked, a RDS survey may be unsuccessful [16]. Unlike our results, findings in the Eastern Caribbean indicated disinterest among MSM in BBS participation, preference for MSM

interviewers, and absence of strong network ties necessary for successful RDS [16]. The subsequent survey in St Vincent and Grenadines, Eastern Caribbean, which used RDS and non-MSM interviewers, was terminated shortly after its commencement due to recruitment challenges [16]. Beyond recruitment and data collection staff, formative results can be used to inform other survey implementation considerations. While monetary and non-monetary compensation was not indicative of survey failure in a quantitative analysis of 128 HIV surveillance surveys among key populations [17], identifying appropriate survey compensation through formative research can help to avoid slow recruitment, recruitment coupon bartering or selling, or misrepresentation of survey eligibility from potential participants. Other areas of input identified in formative assessments can include referral organizations, potential data collections sites and seeds, and considerations for staff selection and training, as demonstrated and applied in Zimbabwe.

There are limitations of this assessment. Participants were recruited in partnership with multiple key population organizations. As a result, MSM/OAMSM participants in the formative assessment may be more comfortable disclosing their sexual behavior or may already have contact with key population-friendly services compared to those who did not participate. As such, the assessment may not represent all perspectives shared by individuals in the MSM/OAMSM communities in Harare and Bulawayo. We sought to overcome this limitation by purposively recruiting participants with diverse demographic characteristics including age and marital status, factors that may affect openness in disclosing someone is a member of the MSM/OAMSM community. Based on stakeholder feedback received after the formative assessment, transgender women were actively recruited to participate in the BBS as purposively selected seeds [15] however since formative assessment tools did not capture information specific to gender identity, acceptability and appropriateness of BBS approaches among transgender women or gender non-confirming individuals assigned male sex at birth were not explicitly explored in this formative assessment. While data presented were not transcribed nor coded, identifying themes from audio recordings allowed for the capturing of non-verbal information such as sighs, delays in response, or utterances of agreement that may otherwise be lost in transcription while also facilitating rapid use of results in programmatic settings. Though focus groups were stratified by age to facilitate participant rapport, data were not analyzed separately by age group given the small sample size. Therefore, we were unable to effectively explore differences between older and younger participants. Despite these limitations, these results are the first to characterize acceptable and appropriate implementation strategies of a BBS among MSM/OAMSM in Zimbabwe and provide insight into factors affecting survey success.

## Conclusions

Findings from this assessment highlight contextually relevant considerations for BBS implementation including behaviors and languages to avoid when working with MSM/OAMSM in urban Zimbabwe, appropriate incentive amounts for survey participation, proposed data collection sites, and differences in sexual openness, marital status, and networks among younger and older MSM/OAMSM. Our MSM/OAMSM participants reported an interest in BBS participation as it would allow them to express themselves, and provide an opportunity for MSM/OAMSM who may otherwise be disengaged in services to access needed HIV counseling and testing. Participants also reported strong network ties suggesting RDS to be an appropriate recruitment method for BBS in this context. Findings from this formative assessment elucidated key considerations and strategies for implementation to ensure the subsequent BBS in urban Zimbabwe was both acceptable and appropriate and can inform future BBS and surveillance efforts in contextually similar countries.

## Supporting information

**S1 Text. IDI guide MSM 25 June 2018.**
(PDF)

**S2 Text. FGD guide 25 June 2018.**
(PDF)

**S3 Text. IDI guide facilitator 25 June 2018.**
(PDF)

**S4 Text. IDI guide service provider 25 June 2018.**
(PDF)

**S5 Text. PLOS inclusivity in global research questionnaire.**
(PDF)

## Acknowledgments

We thank the participants for their time and contribution to the study and staff involved in data collection. We are grateful to the Ministry of Health and Child Care, Gay and Lesbians of Zimbabwe, TranSmart, Population Services International, Pangaea Zimbabwe Aids Trust, Centre for Sexual Health and HIV/AIDS Research Zimbabwe, Hands of Hope, and Sexual Rights Centre for their support in recruitment and staff training.

**Disclaimer:** Lauren Parmley was not at USAID when the research for the current paper was conducted. The views and opinions expressed in this paper are those of the authors and not necessarily the views and opinions of the United States Agency for International Development.

## Author Contributions

**Conceptualization:** Lauren E. Parmley, Tiffany G. Harris, Owen Mugurungi, John H. Rogers, Avi Hakim, Godfrey Musuka, Innocent Chingombe, Munyaradzi Mapingure.

**Data curation:** Lauren E. Parmley, Godfrey Musuka, Innocent Chingombe, Munyaradzi Mapingure.

**Formal analysis:** Lauren E. Parmley, Sophia S. Miller, Munyaradzi Mapingure.

**Funding acquisition:** Tiffany G. Harris, John H. Rogers, Godfrey Musuka, Innocent Chingombe, Munyaradzi Mapingure.

**Investigation:** Lauren E. Parmley, Tiffany G. Harris, Owen Mugurungi, John H. Rogers, Avi Hakim, Godfrey Musuka, Innocent Chingombe, Munyaradzi Mapingure.

**Methodology:** Lauren E. Parmley, Tiffany G. Harris, Owen Mugurungi, John H. Rogers, Avi Hakim, Godfrey Musuka, Innocent Chingombe, Munyaradzi Mapingure.

**Project administration:** Lauren E. Parmley, Tiffany G. Harris, John H. Rogers, Avi Hakim, Godfrey Musuka, Innocent Chingombe, Munyaradzi Mapingure.

**Supervision:** Lauren E. Parmley, Tiffany G. Harris, Owen Mugurungi, John H. Rogers, Avi Hakim, Godfrey Musuka, Innocent Chingombe, Munyaradzi Mapingure.

**Validation:** Lauren E. Parmley, Tiffany G. Harris, Godfrey Musuka, Innocent Chingombe, Munyaradzi Mapingure.

**Writing – original draft:** Lauren E. Parmley, Sophia S. Miller.

**Writing – review & editing:** Lauren E. Parmley, Sophia S. Miller, Tiffany G. Harris, Owen Mugurungi, John H. Rogers, Avi Hakim, Godfrey Musuka, Innocent Chingombe, Munyaradzi Mapingure.

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
