## [Decision Letter · Decision Letter 0]

26 Jul 2022

PGPH-D-21-01054

Characterizing acceptable and appropriate implementation strategies of a biobehavioral survey among men who have sex with men in Zimbabwe

Dear Dr. Parmley,

Thank you for submitting your manuscript to PLOS Global Public Health. After careful consideration, we feel that it has merit but does not fully meet PLOS Global Public Health’s publication criteria as it currently stands. Therefore, we invite you to submit a revised version of the manuscript that addresses the points raised during the review process.

We look forward to receiving your revised manuscript.

Kind regards,

Rachel Hall-Clifford

Academic Editor

Journal Requirements:

1. Please include a complete copy of PLOS’ questionnaire on inclusivity in global research in your revised manuscript. Our policy for research in this area aims to improve transparency in the reporting of research performed outside of researchers’ own country or community. The policy applies to researchers who have travelled to a different country to conduct research, research with Indigenous populations or their lands, and research on cultural artefacts. The questionnaire can also be requested at the journal’s discretion for any other submissions, even if these conditions are not met.  Please find more information on the policy and a link to download a blank copy of the questionnaire here: https://journals.plos.org/plosone/s/best-practices-in-research-reporting. Please upload a completed version of your questionnaire as Supporting Information when you resubmit your manuscript.

Please update the Funding Informatio not matched with Financial Disclosure Statement.

3. Please send a completed 'Competing Interests' statement, including any COIs declared by your co-authors. If you have no competing interests to declare, please state "The authors have declared that no competing interests exist". Otherwise please declare all competing interests beginning with the statement "I have read the journal's policy and the authors of this manuscript have the following competing interests:"

Additional Editor Comments (if provided):

Thank you for your submission. Prior to publication, some revisions are required, including clarity on your methods, particularly inclusion criteria and participants. Please see reviewer comments.

Reviewers' comments:

Reviewer's Responses to Questions

**Comments to the Author**

1. Does this manuscript meet PLOS Global Public Health’s publication criteria? Is the manuscript technically sound, and do the data support the conclusions? The manuscript must describe methodologically and ethically rigorous research with conclusions that are appropriately drawn based on the data presented.

Reviewer #1: Yes

Reviewer #2: Yes

2. Has the statistical analysis been performed appropriately and rigorously?

Reviewer #1: Yes

Reviewer #2: N/A

3. Have the authors made all data underlying the findings in their manuscript fully available (please refer to the Data Availability Statement at the start of the manuscript PDF file)?

Reviewer #1: Yes

Reviewer #2: No

4. Is the manuscript presented in an intelligible fashion and written in standard English?

Reviewer #1: Yes

Reviewer #2: Yes

5. Review Comments to the Author

Reviewer #1: � Thank you for the opportunity to review the manuscript

The paper is well written, and addresses gaps in literature relating to key populations’ behaviour survey which are important in HIV related engagements and interventions for this group

The data and data analysis do support the claims; however, the exclusion criteria have not been sufficiently explained

In light of the fact that same sex relationships are criminalised in Zimbabwe, it is understandable that the informed consent were given verbal. However, the authors need to provide the ethical clearance number.

The paper is well written and is suitable for publication

Reviewer #2: SUMMARY AND OVERALL COMMENTS

This manuscript describes the formative research undertaken prior to the implementation of a respondent driven sampling (RDS) bio-behavioural survey (BBS) among GBMSM in Harare and Bulawayo in Zimbabwe, where a previous BBS among MSM had not until that point been published in a peer-reviewed journal. The approach taken was sound and the findings consistent with the small amount of previous research and programmatic work in Zimbabwe and in other settings sharing some similar contextual characteristics. The authors make clear how their findings informed survey design. As the authors mention, the findings from similar qualitative formative work conducted to inform RDS survey design are often not published or reported in much detail, so it is a helpful contribution to the literature and to the implementation of future research to do so. The paper is nicely written and well-organised.

There were a few questions I had remaining after reading the article:

- were the researchers explicitly aiming to recruit cisgender MSM only? Please clarify and if not, please clarify the extent to which participants included diverse gender identities. There is mention of this in the discussion but I think it needs to be clear what the intended population to recruit was earlier on and I still easn't clear what the original intentions of the survey were.

- to what extent were the FGD and IDI participants (not key informants) representative of MSM who were not engaged with MSM community groups/service providers/prior research already?

- what did the authors find regarding the existence of sub-communities likely present within the social network and the characteristics by which they varied?

A key point was that I was curious about was whether, on reflection, there were any topics NOT covered by the formative research that the researchers later realised would have been useful to cover. I think the main value of this article is in providing a worked example of doing such formative research prior to implementation of a a similar survey in a similar population, so it would be useful to know this.

I have some comments, including minor typos/edits suggested for each section below.

ABSTRACT:

Suggest re-phrasing to avoid repeated word: “Yet the marginalized nature of populations participating in a BBS requires contextually acceptable and appropriate implementation strategies for effective implementation.”

INTRODUCTION

Line 63: suggest ‘about’ MSM instead of ‘on MSM’.

METHODS:

Were there any eligible potential participants who declined to take part in the study?

Holding focus groups separately for older and younger MSM makes sense to me. Were there any other key aspects that researchers might have considered important such as sexual orientation, gender identity or known HIV status that researchers might have considered separate groups for?

Were any other survey designs other than RDS explicitly discussed with FGD and IDI participants, eg venue-based methods? I’m not suggesting that this necessarily would have been more appropriate but I was interest interested in the breadth of what was considered and discussed and reasons why other survey designs might not have been appropriate.

In the spirit of providing a useful resource to other future projects, it would be helpful to attached topic guides and/or other data collection instruments used for this study as an appendix.

RESULTS

Line 169: Here do you mean that the individual interviewees had been involved in service provision for a median of 2 years or that the organisations had only existed this long?

I would find it helpful to better understand the extent to which interviewees and FGD participants were already in touch with LGBT community groups, or service provision. I understand that not all participants were identified by service providers and that snowball sampling was used, but am still curious about how representative the participants might have been of MSM not already served by community groups and clinics. I wondered about this particularly as it was reported Line 202 that many MSM participants had prior experience with HIV testing in research and clinical settings (though I suppose the latter would surprise me less). Also Line 333, where it is implied that some some number had previously been compensated in the context of public health programmes. I see that the authors have commented on this in the Discussion , which is good, but was there any information about this collected from participants that could inform this question?

On compensation, was any separate data collected on compensation in the context of onwards recruitment for the RDS surveys (as these surveys typically compensate participants for both their own participation and for onwards recruitment)?

Were there any differences reported in the extent to which younger and older MSM might be well networked or not?

Was there any information collected about sub-communities that might be expressed as sub-networks for the RDS recruitment, and if so, what was found?

Application to survey methodology- this section was reported quite positively and it does indeed sound like the formative research was useful and important to have done. I was curious though whether there were any topics that in hindsight the researchers wish they had had more data to inform that they could have collected at the formative stage?

DISCUSSION

The research conducted (even if unsuccessfully) among MSM in the Caribbean needs a reference of some kind.

Line 450: It wasn’t clear to me from what was said why issues specific to transgender women were not captured when they were included in the IBBS. Was their inclusion recommended only after completion of the formative research? It might also be helpful to clarify the gender identity of participants and the gender criteria for the final study earlier on because I was wondering about this while reading the paper and it is not mentioned until the discussion.

6. PLOS authors have the option to publish the peer review history of their article (what does this mean?). If published, this will include your full peer review and any attached files.

**Do you want your identity to be public for this peer review?** For information about this choice, including consent withdrawal, please see our Privacy Policy.

Reviewer #1: No

Reviewer #2: **Yes: **Elizabeth Fearon

---

## [Decision Letter · Decision Letter 1]

21 Sep 2022

Characterizing acceptable and appropriate implementation strategies of a biobehavioral survey among men who have sex with men and others assigned male who have sex with men in Zimbabwe

PGPH-D-21-01054R1

Dear Miss Parmley,

We are pleased to inform you that your manuscript 'Characterizing acceptable and appropriate implementation strategies of a biobehavioral survey among men who have sex with men and others assigned male who have sex with men in Zimbabwe' has been provisionally accepted for publication in PLOS Global Public Health.

Best regards,

Rachel Hall-Clifford

Academic Editor

Reviewer Comments (if any, and for reference):

Reviewer's Responses to Questions

**Comments to the Author**

1. If the authors have adequately addressed your comments raised in a previous round of review and you feel that this manuscript is now acceptable for publication, you may indicate that here to bypass the “Comments to the Author” section, enter your conflict of interest statement in the “Confidential to Editor” section, and submit your "Accept" recommendation.

Reviewer #2: All comments have been addressed

2. Does this manuscript meet PLOS Global Public Health’s publication criteria? Is the manuscript technically sound, and do the data support the conclusions? The manuscript must describe methodologically and ethically rigorous research with conclusions that are appropriately drawn based on the data presented.

Reviewer #2: (No Response)

3. Has the statistical analysis been performed appropriately and rigorously?

Reviewer #2: (No Response)

4. Have the authors made all data underlying the findings in their manuscript fully available (please refer to the Data Availability Statement at the start of the manuscript PDF file)?

Reviewer #2: (No Response)

5. Is the manuscript presented in an intelligible fashion and written in standard English?

Reviewer #2: (No Response)

6. Review Comments to the Author

Reviewer #2: (No Response)

7. PLOS authors have the option to publish the peer review history of their article (what does this mean?). If published, this will include your full peer review and any attached files.

**Do you want your identity to be public for this peer review?** For information about this choice, including consent withdrawal, please see our Privacy Policy.

Reviewer #2: **Yes: **Elizabeth Fearon
